# COVID-19 Severity Potentially Modulated by Cardiovascular-Disease-Associated Immune Dysregulation

**DOI:** 10.3390/v13061018

**Published:** 2021-05-28

**Authors:** Abby C. Lee, Grant Castaneda, Wei Tse Li, Chengyu Chen, Neil Shende, Jaideep Chakladar, Pam R. Taub, Eric Y. Chang, Weg M. Ongkeko

**Affiliations:** 1Department of Surgery, Division of Otolaryngology-Head and Neck Surgery, UC San Diego School of Medicine, San Diego, CA 92093, USA; acl008@ucsd.edu (A.C.L.); gecastan@ucsd.edu (G.C.); wtl008@ucsd.edu (W.T.L.); chc401@ucsd.edu (C.C.); nshende@ucsd.edu (N.S.); jchaklad@ucsd.edu (J.C.); 2Research Service, VA San Diego Healthcare System, San Diego, CA 92161, USA; e8chang@health.ucsd.edu; 3Department of Medicine, Division of Cardiovascular Medicine, University of California, San Diego, CA 92093, USA; ptaub@health.ucsd.edu; 4Department of Radiology, University of California, San Diego, CA 92093, USA

**Keywords:** COVID-19, coronary artery disease, cardiomyopathy, venous thromboembolism event, inflammation

## Abstract

Patients with underlying cardiovascular conditions are particularly vulnerable to severe COVID-19. In this project, we aimed to characterize similarities in dysregulated immune pathways between COVID-19 patients and patients with cardiomyopathy, venous thromboembolism (VTE), or coronary artery disease (CAD). We hypothesized that these similarly dysregulated pathways may be critical to how cardiovascular diseases (CVDs) exacerbate COVID-19. To evaluate immune dysregulation in different diseases, we used four separate datasets, including RNA-sequencing data from human left ventricular cardiac muscle samples of patients with dilated or ischemic cardiomyopathy and healthy controls; RNA-sequencing data of whole blood samples from patients with single or recurrent event VTE and healthy controls; RNA-sequencing data of human peripheral blood mononuclear cells (PBMCs) from patients with and without obstructive CAD; and RNA-sequencing data of platelets from COVID-19 subjects and healthy controls. We found similar immune dysregulation profiles between patients with CVDs and COVID-19 patients. Interestingly, cardiomyopathy patients display the most similar immune landscape to COVID-19 patients. Additionally, COVID-19 patients experience greater upregulation of cytokine- and inflammasome-related genes than patients with CVDs. In all, patients with CVDs have a significant overlap of cytokine- and inflammasome-related gene expression profiles with that of COVID-19 patients, possibly explaining their greater vulnerability to severe COVID-19.

## 1. Introduction

In December 2019, widespread infection by severe acute respiratory syndrome coronavirus 2 (SARS-CoV-2) was reported in Wuhan, China [1]. Since then, SARS-CoV-2, which causes COVID-19, has spread rapidly, and COVID-19 was declared a pandemic by the World Health Organization (WHO) on 11 March 2020 [2] Current research suggests that patients with existing comorbidities, including hypertension, cardiovascular disease, diabetes, and obesity are more likely to develop severe COVID-19 [3,4,5]. COVID-19 has also been known to induce myocardial injury, arrhythmia, acute coronary syndrome, and venous thromboembolism (VTE) [6,7,8]. Such cardiovascular damage has been attributed to cytokine storms triggered by the SARS-CoV-2 infection that can cause multi-organ damage [9,10]. Additionally, COVID-19 patients experience coagulation abnormalities, possibly leading to an increased risk of thromboembolic events [11]. In multiple autopsy studies, thromboembolic events were identified in patients who had COVID-19 [12,13]. Specifically, in Schurink et al., it was found in multiple organs, including but not limited to the brain, lungs, heart, and kidneys [13]. Undeniably, research suggests links between cardiovascular disease (CVD) and COVID-19. However, the mechanisms by which CVD results in poorer COVID-19 prognosis remains unclear. As CVD encompasses a wide range of specific disorders, it would be impractical to obtain a dataset for all these disorders. In this study, we focused on three of the most common cardiovascular conditions: cardiomyopathy, VTE, and CAD.

Cardiomyopathy refers to diseases of the myocardium associated with mechanical and/or electricdysfunction [14]. In nonischemic dilated cardiomyopathy (NIDCM), the heart’s ventricles are enlarged [15]. Cytokines and inflammasomes are known to play significant roles in cardiomyopathy pathogenesis, which suggests a commonality between cardiomyopathy and COVID-19, where excess inflammation is often induced [16,17,18].

VTE includes deep vein thrombosis, where a blood clot forms in a deep vein, typically in the lower extremities or pelvis, which may dislodge and result in pulmonary embolism (PE). Similar to VTE, COVID-19 patients have increased oxidative stress, which is one of the hallmarks for endothelial damage [19,20,21]. Additionally, COVID-19 patients have been shown to be at risk of thrombotic events [22,23,24]. Interestingly, in COVID-19 patients with thrombotic events, their D-dimer levels were found to be increased [23,24]. It is well established that the immune system functions in deep vein thrombosis pathogenesis, and the restriction of venous blood flow leads to the recruitment of neutrophils, monocytes, and platelets [25,26,27]. Since higher levels of monocytes and neutrophils have been observed in COVID-19 patients requiring ICU hospitalization, it is possible that such pre-existing immune dysregulation in COVID-19 VTE patients increases their risk of progressing to severe disease [28,29,30].

Lastly, coronary artery disease (CAD) pathogenesis also has an established immunological component [31]. Higher levels of C-reactive protein (CRP) [32], leukocytes [33], and cytokines [34] are associated with both CAD and severe COVID-19 patients [35,36,37]. Moreover, excessive pro-inflammatory cytokine production is associated with vascular damage that induces uncontrolled blood clotting [38]. This not only suggests that CAD patients are more vulnerable to severe COVID-19 [39], but also suggests that COVID-19 may exacerbate CAD.

In this project, we aimed to characterize and compare the dysregulation of the immune landscape in patients with cardiomyopathy, VTE, CAD, and COVID-19. We analyzed the expression of cytokine genes and inflammasome-related genes, the extent of immune infiltration, and the enrichment of immunological pathways and signatures. By comparing these features of the immune system, we hoped to gain a more comprehensive understanding of the cardiovascular-disease-mediated immune dysregulation that leaves patients more vulnerable to severe COVID-19.

## 2. Materials and Methods

### 2.1. Downloading Data

RNA-sequencing data were obtained from the following datasets: GSE116250 [40], GSE19151 [41], GSE90074 [42], and SRP262885 [43]. GSE116250, provided by Sweet et al., consists of the RNA sequencing of human left ventricular samples from 14 patients with no major cardiac history (nonfailing), 37 patients with NIDCM, and 13 patients with ICM. GSE19151, provided by Lewis et al., consists of the high-throughput sequencing of whole blood samples from 63 healthy controls, 23 patients with single VTE, and 17 patients with recurrent VTE on warfarin. GSE90074 consists of the RNA-sequencing data of PBMCs from 93 patients with and 50 patients without CAD. Lastly, SRP262885 consists of the RNA-sequencing data of platelets from 10 COVID-19 subjects and 5 healthy controls.

### 2.2. Differential Expression

For the cardiomyopathy and VTE cohorts, the Kruskal–Wallis test (*p* < 0.05) was used to determine differentially expressed genes. CAD cohorts were analyzed using the GEO2R software, which employs the limma (linear models for microarray analysis) R package (*p* < 0.05). Differential expression was applied to the COVID-19 platelet data to determine the genes that were differentially expressed (*p* < 0.05).

### 2.3. GSEA

To correlate gene expression to immune-associated signatures, gene set enrichment analysis (GSEA) was utilized. Pathways were chosen from the C2: CP set of signatures from the Molecular Signatures Database [44]. Signatures that were significantly enriched were those with a nominal *p*-value < 0.05.

### 2.4. CIBERSORTx

The CIBERSORTx algorithm was used to deconvolute the RNA-sequencing data to estimate the infiltration levels of 22 immune cell types [45].

## 3. Results

### 3.1. Comparing Immune Profiles of COVID-19 and Cardiomyopathy Patients

#### 3.1.1. Similarities in Immune-Associated Gene Dysregulation in COVID-19 and Cardiomyopathy

Gene dysregulation was determined by comparing COVID-19 and cardiomyopathy samples to healthy controls for each study. Cardiomyopathy samples were separated into patients with ischemic cardiomyopathy (ICM) or nonischemic dilated cardiomyopathy (NIDCM). The two groups were individually compared against samples from patients with no major cardiovascular disease (healthy controls).

We found a significant overlap between COVID-19 patients and ICM and NIDCM patients’ immune-associated (IA) gene expression. About half of the IA genes dysregulated in COVID-19 are dysregulated in either or both types of cardiomyopathy (Figure 1A). A complete list of dysregulated IA genes are found in Appendix A, Table A1. Cytokine-related genes that are dysregulated in both cardiomyopathy patients and COVID-19 patients include chemokines (CCL3, CCL4, CXCL4, etc.), interleukins or interleukin receptors (IL15, IL20RA, etc.), and genes in the transforming growth factor beta (TGFB) family. The inflammasome-related genes include genes in the caspase family (CASP2, CASP9, etc.), mitogen-activated protein kinase (MAPK)-related genes, and nuclear factor-kB (NF-kB) regulators (IKBKG, NFKBIA, etc.). IA gene dysregulation was very similar between dilated and ischemic cardiomyopathies. We observed that a significant number of IA genes were dysregulated in either of the cardiomyopathies but not in COVID-19 (Figure 1A).

Interestingly, we found that most of the genes dysregulated in both COVID-19 and cardiomyopathy were dysregulated to a greater degree in COVID-19 than in cardiomyopathy samples. This was observed for TGFB3, CCL4, IL15, and IL20RA in both ICM samples vs. COVID-19 samples and NIDCM samples vs. COVID-19 samples (Figure 1B). Furthermore, these dysregulated genes appeared to be similarly dysregulated in COVID-19 and corresponding healthy samples (Figure 1C). In contrast, these genes’ expression in cardiomyopathy samples and corresponding healthy samples were only sometimes similar, without overwhelming differences in expression levels between the two cohorts (Figure 1C). Therefore, we believe that these dysregulated genes are dysregulated to a greater degree in COVID-19 than in cardiomyopathy.

The inflammasome-associated genes dysregulated in both COVID-19 and cardiomyopathy were upregulated in both conditions (Figure 1C), suggesting that they may upregulate inflammation through similar pathways.

#### 3.1.2. Comparison of Immune Cell Population Abundance in COVID-19 vs. Cardiomyopathy

We discovered that the levels of T and B cells were unchanged in healthy vs. COVID-19 patients (See Appendix A, Figure A1A). The most noticeable change in immune cell abundance occurred in macrophages for COVID-19 patients, where M0 macrophage levels were dramatically reduced and M1 and M2 macrophage levels were slightly increased (See Appendix A, Figure A1A). Both cardiomyopathies elicited greater immune cell abundance changes than COVID-19, with the changes being more pronounced for ICM. The levels of M1 and M2 macrophages increased in ICM, similar to what was observed for COVID-19 (See Appendix A, Figure A1B). The levels of T- and B-cell subtypes changed more dramatically in ICM and NIDCM than in COVID-19. In summary, the levels of inflammatory macrophages increased for both cardiomyopathy and COVID-19 patients, while the levels of other immune cell types did not correlate between the two conditions.

#### 3.1.3. Evaluation of Canonical Pathways Correlated with Genes Dysregulated in Both COVID-19 and Cardiomyopathy

We analyzed genes that are dysregulated in both COVID-19 and cardiomyopathy to assess if they dysregulate common pathways in the two conditions. Interleukin 1 receptor-associated kinase 2 (IRAK2), upregulated in both COVID-19 and ICM, was associated with the upregulation of the FCER1 and TP63 pathways, both of which are associated with inflammation and immune activation (See Appendix A, Figure A2A) [46,47]. IRAK2 is a promoter of NF-kB signaling [48]. Caspase 2 (CASP2), also upregulated in COVID-19 and cardiomyopathy, is associated with the downregulation of IFIH, which is capable of recognizing viruses and inducing inflammation [49,50]. Finally, CYLD lysine 63 deubiquitinase (CYLD) was correlated with multiple identical pathways for both COVID-19 samples and ICM samples. CYLD is upregulated in both COVID-19 and cardiomyopathy and was found to correlate with the activation of FGFR2, an important promoter of inflammation [51], and TXA2, a gene that is upregulated in platelets (See Appendix A, Figure A2A) [52]. CYLD is an inhibitor of inflammation [53]. Since the majority of correlations were between IA genes and pro-inflammatory pathways and signatures, the dysregulation of CYLD represents an exception, and we hypothesize that CYLD may be expressed as a response to attenuate excessive inflammation. We found that the overwhelming majority of pathways that correlated with dysregulated genes in both COVID-19 and NIDCM are associated with CYLD, and these pathways are primarily pro-plotting, pro-cell aggregation, and pro-inflammation (See Appendix A, Figure A2A), supporting the possibility that CYLD is released in response to inflammation.

### 3.2. Comparing Immune Profiles of COVID-19 and VTE Patients

#### 3.2.1. Similarities in Immune-Associated Gene Dysregulation in COVID-19 and VTE

We compared the immune landscape between COVID-19 samples and blood samples from VTE patients to find similarities in IA gene and pathway expression. VTE patients were classified into single occurrence VTE (single VTE) and recurrent VTE. Compared to the similarities in IA genes dysregulated between COVID-19 and cardiomyopathy, the similarities between COVID-19 and VTE are less pronounced.

Two cytokine-associated genes (CCL4 and CD40) were dysregulated in COVID-19 and single VTE, and one cytokine-associated gene (CCL4) was dysregulated in COVID-19 and recurrent VTE (Figure 2A). A complete list of dysregulated cytokine- and inflammasome-associated genes are found in Appendix A, Table A2. CCL4 recruits immune cells, including macrophages, monocytes, and T cells [54], suggesting that COVID-19 and VTE may both exhibit the increased recruitment of inflammatory immune cells. The upregulation of CCL4 was much greater in COVID-19 than in VTE, however (Figure 2B). Two inflammasome-related genes were found to be dysregulated in VTE and COVID-19. BCL2L1 is known to be highly upregulated in inflamed tissue [55], and it was found to be upregulated in COVID-19 and both single and recurrent VTE (Figure 2C), while CASP4 directs the noncanonical upregulation of inflammasomes [56]. Interestingly, CASP4 was found to be upregulated in both COVID-19 and recurrent VTE but downregulated in single VTE (Figure 2C), which suggests that the gene could contribute to the development of recurrent VTE.

#### 3.2.2. Comparison of Immune Cell Population Abundance in COVID-19 vs. VTE

We found that naive B cells were dramatically reduced in abundance in VTE patients, which may indicate adaptive immune activation (See Appendix A, Figure A1C). This was the only significant immune cell population change in VTE patients that was observed and does not correlate to changes in COVID-19.

#### 3.2.3. Evaluation of Canonical Pathways Correlated with Genes Dysregulated in COVID-19 and VTE

BCL2L1 and CASP4 were the only genes found to be dysregulated in both COVID-19 and VTE and also correlated with similar pathways in both patient cohorts (Figure 3A,B). BCL2L1 was found to be upregulated in COVID-19 and both VTE cohorts (Figure 3C). However, the direction of correlation to pathways was the complete opposite between COVID-19 and recurrent VTE (See Appendix A, Figure A2C). The high correlation strength for each cohort suggests BCL2L1 is involved in both COVID-19 and recurrent VTE but functions in opposite ways. On the other hand, CASP4 expression was correlated with over 10 pathways in the same direction for both COVID-19 and recurrent VTE (See Appendix A, Figure A2C). It was also found to be upregulated in both COVID-19 and recurrent VTE (Figure 3C). The pathways correlated with CASP4 were immune related (antigen processing and cross presentation) and general metabolism related (ABC transporter, oxidative phosphorylation). Therefore, while CASP4 likely functions similarly in COVID-19 and recurrent VTE, its precise role requires further investigation.

### 3.3. Comparing Immune Profiles of COVID-19 and CAD Patients

#### 3.3.1. Similarities in Immune-Associated Gene Dysregulation in COVID-19 and CAD

We found a significant overlap in IA gene expression in COVID-19 and CAD. About a third of the IA genes dysregulated in CAD were also found to be dysregulated in COVID-19 (Figure 4A). The complete list of dysregulated cytokine and inflammasome-associated genes are found in Appendix A, Table A3.

Cytokine-related genes that were found to be dysregulated in both CAD patients and COVID-19 patients include chemokines (CCL3 and CCL4), chemokine receptor CXCR1, and a TNFSF gene, LTB. The inflammasome-related genes that were found include NF-kB regulators (NFKBIA and CHUK), an alpha arrestin (TXNIP), and an F-BAR domain-containing protein (PSTPIP1). Similar to cardiomyopathy, we found that most genes dysregulated in both COVID-19 and CAD were dysregulated to a greater degree in COVID-19 samples than in CAD samples. This was observed for CCL3, CCL4, CXCR1, and LTB (Figure 4B).

#### 3.3.2. Comparison of Immune Cell Population Abundance in COVID-19 vs. CAD

Similar to COVID-19 patients, the memory B cells in CAD patients were more abundant (See Appendix A, Figure A1D).

#### 3.3.3. Evaluation of Pathways Correlated with Genes Dysregulated in COVID-19 and CAD

We analyzed genes that are dysregulated in both COVID-19 and CAD to assess if they are associated with similar pathways in the two conditions. Notably, we discovered that CHUK, PSTPIP1, and CCL3, upregulated in both COVID-19 and CAD, were associated with the upregulation of many inflammatory pathways in both conditions, including the IL12, IL10, IL23, and P53 regulation pathways (See Appendix A, Figure A2C).

## 4. Discussion

In this project, we characterized the immune landscape of cardiomyopathy, VTE, CAD, and COVID-19 patients, drawing important parallels between COVID-19 and cardiovascular-disease-mediated immune dysregulation. Of the four genes that were more severely dysregulated in COVID-19 compared to cardiomyopathy, two were reported to be dysregulated in COVID-19 patients: pro-inflammatory CCL4 was highly expressed in the bronchoalveolar lavage fluid of COVID-19 patients [57], and IL15 modulates inflammation and functions in viral clearance [58,59]. In fact, IL15 is part of an immune-based biomarker signature associated with mortality in COVID-19 patients, and CCL4 has been shown to be elevated in COVID-19 patients who eventually died due to the disease [60]. As we report that these cytokines are also upregulated in patients with cardiomyopathy, it is possible that such pre-existing immune dysregulation could explain the higher COVID-19 mortality rates of patients with cardiomyopathy and COVID-19. Our findings on immune cell abundance in COVID-19 and cardiomyopathy patients also point to a more robust innate immune response in COVID-19 patients, which is plausible as research has shown that a hyperinflammatory innate immune response coupled with an impaired adaptive immune response may lead to tissue damage in COVID-19 patients [59,60,61]. Conversely, the elevated levels of T and B cells in cardiomyopathy patients indicate a stronger adaptive immune response, which is now considered an increasingly important factor in cardiovascular disease pathogenesis [62,63,64]. Comparing COVID-19 and cardiomyopathy patients, we found elevated levels of inflammatory macrophages in both groups of patients. This could suggest that cardiomyopathy patients are more susceptible to hyperinflammation in COVID-19 and are thus more likely to progress to severe COVID-19.

We then analyzed overlapping gene expression pathway dysregulation between cardiomyopathy and COVID-19 patients. Upregulated CASP2 and IRAK2 are of particular interest due to their inflammatory roles. CASP2 is a pro-inflammatory gene and IRAK2 promotes NF-kB, which is a central activator of inflammation. Overall, genes dysregulated in both cardiomyopathy and COVID-19 appear to promote inflammation, which may indicate why cardiovascular disease patients experience poorer clinical outcomes, as greater inflammation correlates with severity and death in COVID-19 [56].

Exploring immune-associated (IA) gene dysregulation in VTE and COVID-19 revealed several IA genes dysregulated in both conditions. Cytokine CCL4 has been shown to be upregulated in COVID-19 patients [65] and in patients who develop cardiovascular diseases [66]. Inflammasome-related genes BCL2L1 and CASP4 are tied to inflammatory caspases. CASP4 is an inflammatory caspase and promotes pro-inflammatory cytokine secretion [67]. Conversely, BCL2L1 inhibits caspase release. With both genes being upregulated in VTE and COVID-19, future analysis must be carried out to examine how these genes function differently in VTE and COVID-19. Interestingly, in both COVID-19 and single VTE, BCL2L1 expression is negatively correlated to canonical pathway expression, but in recurring VTE they are positively correlated. CASP4, on the other hand, only has overlapping significant canonical pathways with COVID-19 in recurrent VTE. From these pathways, we observe that CASP4 functions similarly in recurrent VTE and COVID-19. Together, these results show that VTE and COVID-19 patients share similar upregulation of inflammation-associated genes, which could explain why rates of venous thromboembolism events are higher in COVID-19 patients, as well as why venous thromboembolism events are associated with higher risk of death in COVID-19 patients [68,69].

Lastly, we compared the immune landscape and canonical pathways of CAD and COVID-19 patients. Of the significantly dysregulated cytokines in both COVID-19 and CAD, pro-inflammatory cytokines CCL3 and CCL4 have been associated with COVID-19 severity [59]. Of the inflammasome-related genes, CHUK is of particular interest. CHUK forms part of the IκB kinase (IKK) complex that is involved in the phosphorylation and degradation of IκBα, allowing for the transcription of NF-kB-dependent genes. Following coronavirus infection, the NF-kB pathway is activated via the MyD88 pathway [70], and increased transcription of NF-kB-dependent genes has implications for cardiomyopathy, atherosclerosis, and COVID-19 severity. Specifically, NF-kB activation in endothelial cells triggers the expression of adhesion molecules that are responsible for the invasion and homing of macrophages [71,72,73,74,75,76], contributing to atherosclerosis pathogenesis [77]. In addition, TNFa and IL6 expressions have been shown to be triggered by SARS via the NF-kB pathway [78]. These cytokines have been implicated in macrophage activation syndrome and cytokine storms and are associated with COVID-19 severity [79,80,81]. Interestingly, IRAK2, another NF-kB pathway regulator, is upregulated in cardiomyopathy patients. IRAK2, when phosphorylated with IRAK1 and IRAK4, recruits Ub ligase and activates TRAF6. TRAF6 activates the NF-kB pathway via the IKK complex. In summary, COVID-19 upregulates both IRAK2 and CHUK, while atherosclerosis only upregulates CHUK, and cardiomyopathy upregulates IRAK2, suggesting that NF-kB activation may be critical in all three conditions. Given that hyperactivation of the NF-kB pathway in B cells has been implicated in cytokine storms and the pathogenesis of severe and critical COVID-19 [82], our results suggest that the upregulation of this pathway in patients with pre-existing cardiovascular disease could be key to explaining their poorer COVID-19 prognoses.

## 5. Conclusions

In conclusion, we found that cardiomyopathy, VTE, and CAD patients display significant similarities in inflammation-related gene expression to COVID-19 patients. Therefore, when a patient with the above cardiovascular conditions contracts COVID-19, COVID-19 could further dysregulate the expression of inflammatory genes already dysregulated, leading to more severe inflammation. This may explain why patients with cardiovascular disease are more likely to develop severe COVID-19 and tend to have poorer clinical outcomes [83,84]. Furthermore, we found that patients with CAD display a similar dysregulated immune landscape to COVID-19 patients, possibly indicating why CAD patients are at higher risk of severe COVID-19. Interestingly, cardiomyopathy patients display more similar immune dysregulation to COVID-19 patients than VTE or CAD patients vs. COVID-19 patients. This observation could explain the fact that COVID-19 mortality is increased in congestive heart failure patients, as demonstrated in a study of 31,461 adults [85]. Our findings suggest that investigating the relationships between specific cardiovascular diseases and COVID-19 severity and mortality is meaningful and offers insight into COVID-19 immune dysregulation. However, our study has several limitations. We had limited COVID-19 platelet data, specifically normal patients. This may have impacted our differential expression analysis and thus reduced the statistical power of our analysis. However, the direction of dysregulation of many of the genes identified was consistent with existing literature. Additionally, we used platelet data instead of blood samples. To validate our results, in vitro and in vivo experiments can be done. Despite these limitations, we believe our study advances our understanding of the relationship between cardiovascular disease and COVID-19. Our study also encourages the examination of potential treatment strategies such as anti-inflammatory steroids and ACE2 inhibitors to downregulate inflammation in COVID-19 patients with CVDs.

## Figures and Tables

**Figure 1 viruses-13-01018-f001:**
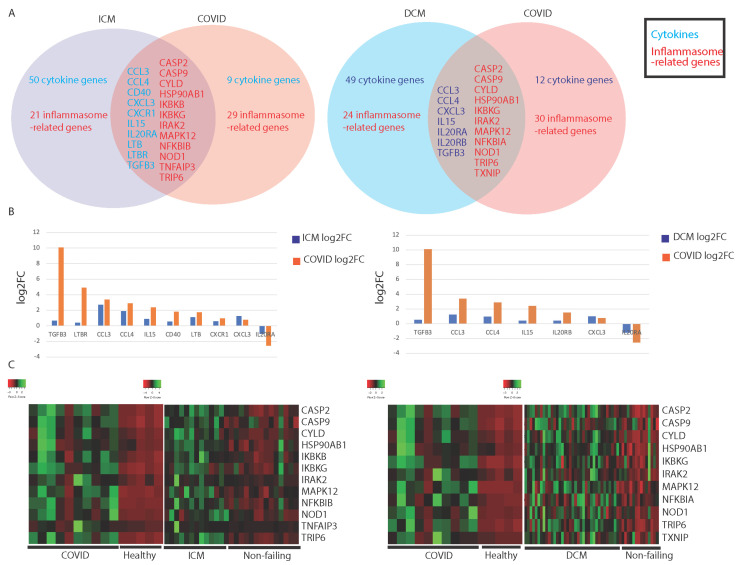
Comparing ischemic cardiomyopathy (ICM), nonischemic dilated cardiomyopathy (NIDCM), and COVID-19 patients. (**A**) Summary of commonly dysregulated cytokine- and inflammasome-related genes in COVID-19 and ICM/NIDCM patients. Cytokines are represented in blue and inflammasome-related genes are in red. (**B**) Bar plots of the log2 fold change of significantly dysregulated cytokine genes in COVID-19 and ICM/NIDCM patients. (**C**) Heatmaps illustrating similar patterns of dysregulation of inflammasome-related genes in COVID-19 and ICM/NIDCM patients compared to their respective controls.

**Figure 2 viruses-13-01018-f002:**
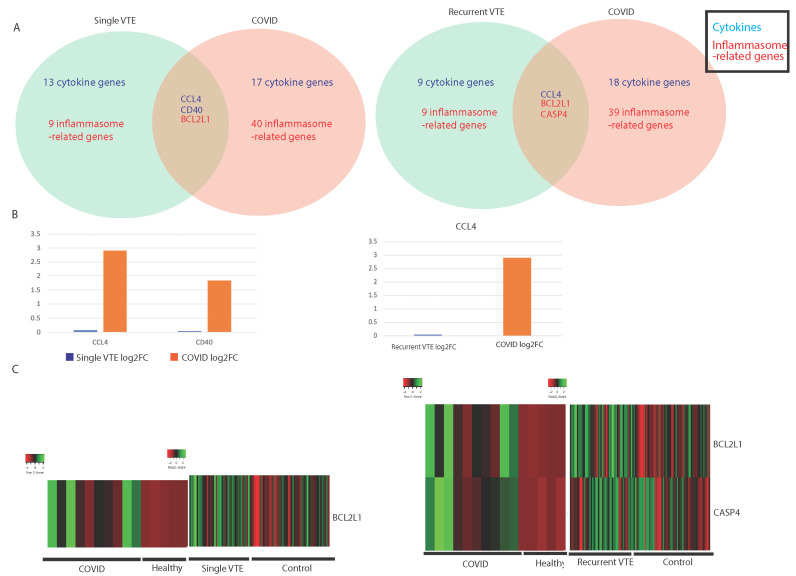
Comparing single venous thromboembolism (VTE), recurrent VTE, and COVID-19 patients. (**A**) Summary of commonly dysregulated cytokine- and inflammasome-related genes in COVID-19 and single/recurrent VTE patients. Cytokines are denoted in blue and inflammasome-related genes are in red. (**B**) Bar plots of the log2 (fold change) of significantly dysregulated cytokine genes in COVID-19 and single/recurrent VTE patients. (**C**) Heatmaps of inflammasome-related genes in COVID-19 and single/recurrent VTE patients.

**Figure 3 viruses-13-01018-f003:**
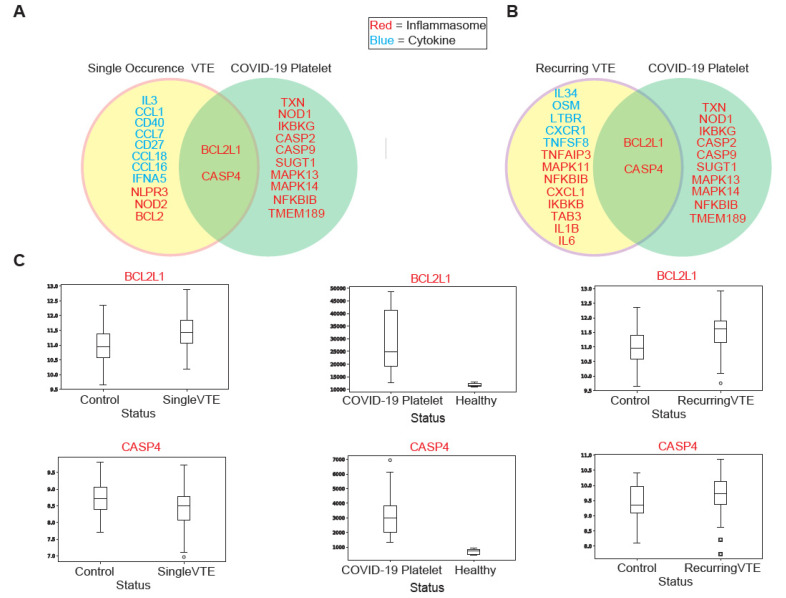
(**A**) Summary of common dysregulated genes correlated with pathways in single venous thromboembolism (VTE) and COVID-19. (**B**) Summary of common dysregulated genes correlated with pathways in recurring VTE and COVID-19. (**C**) Boxplots of CASP4 and BCL2L1 expression in COVID-19 and VTE cohorts.

**Figure 4 viruses-13-01018-f004:**
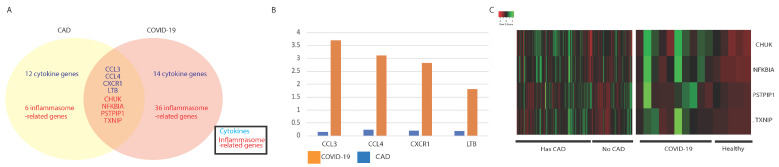
Comparing coronary artery disease (CAD) and COVID-19 patients. (**A**) Summary of commonly dysregulated cytokine- and inflammasome-related genes in COVID-19 and CAD patients. Cytokines are in blue and inflammasome-related genes are in red. (**B**) Bar plots of the log2 fold change of significantly dysregulated cytokine genes in COVID-19 and CAD patients. (**C**) Heatmaps of inflammasome-related genes in COVID-19 and CAD patients.

## Data Availability

The data can be found in the following studies: (1) GSE116250: Sweet ME, Cocciolo A, and Slavov D et al. Transcriptome analysis of human heart failure reveals dysregulated cell adhesion in dilated cardiomyopathy and activated immune pathways in ischemic heart failure. *BMC Genom.*
**2018**, *19*, 812, doi:10.1186/s12864-018-5213-9; (2) GSE19151: Lewis DA, Stashenko GJ, and Akay OM et al. Whole blood gene expression analyses in patients with single versus recurrent venous thromboembolism. *Thromb. Res.*
**2011**, *128*, 536–540, doi:10.1016/j.thromres.2011.06.003; (3) GSE90074: Ravi S SR, Hilliard E, and Lee CR. Clinical Evidence Supports a Protective Role for CXCL5 in Coronary Artery Disease. *Am. J. Pathol.*
**2017**, *187*, 2895–2911, doi:10.1016/j.ajpath.2017.08.006; (4) SRP262885: RNA-seq of platelets from SARS-CoV-2 COVID-19. University of Utah, **2020**.

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
