# Peer review of "COVID-19 Severity Potentially Modulated by Cardiovascular-Disease-Associated Immune Dysregulation"

_viruses, 2021, doi:10.3390/v13061018_

Round 1

Reviewer 1 Report

well written study worthy to be published

Author Response

We thank the reviewer for their valuable feedback, and are glad they found our results useful as well.

Reviewer 2 Report

Authors present an interesting comparison of cardiovascular diseases (VTE, CAD, cardiomyopathy)-related genes expression and COVID-19 related genes expression, leading to some similarities and trying to explain cardiovascular system involvement in COVID-19.

However:

1) please do not generalize that cardiomyopathy is an inflammation of heart muscle, there are many other causes of heart damage and inflammation is not the main trigger (Miyazawa K, Ito K. The Evolving Story in the Genetic Analysis for Heart Failure. Front Cardiovasc Med. 2021 Apr 13;8:646816. doi: 10.3389/fcvm.2021.646816. eCollection 2021).

2) please try to be more specific than cloths are formed in a deep vein (which?)

3) please do not use obvious information that increased D-dimer levels may indicate there is a significant blood clot (line 62),

4) it is not necessary to use word 'obstructive' with CAD,

5) please if possible improve the resolution of all Figures,

6) please do not use so many agents abbreviations without explaining them, especially in Results section, it can be hard for Reader to remember the significance of your results,

7) please try to be more specific in the discussion section about significance of your data, if possible add more clinical data.

Round 2

Reviewer 2 Report

Authors answered to all my questions sufficiently, the manuscript is much clearer rught now.